# Enhanced Catalytic Activity of TEMPO-Mediated Aerobic Oxidation of Alcohols via Redox-Active Metal–Organic Framework Nodes

**DOI:** 10.3390/molecules28020593

**Published:** 2023-01-06

**Authors:** Bing Wang, Junjie Zhang, Yun Xue, Yuliang Chong, Dongdong Zhao, Hu Cheng, Liangliang Tian, Jinliang Zhuang

**Affiliations:** 1Key Lab for Functional Materials Chemistry of Guizhou Province, School of Chemistry and Materials Science, Guizhou Normal University, Guiyang 550001, China; 2School of Electronic Information and Electrical Engineering, Chongqing University of Arts and Sciences, Chongqing 400000, China

**Keywords:** metal–organic frameworks, TEMPO radical, alcohol oxidation, synergistic effect, redox

## Abstract

Metal–organic frameworks (MOFs) are outstanding platforms for heterogeneous catalysis due to their tunable pore size, huge surface area, large porosity, and potential active sites. The design and synthesis of MOF/organocatalyst co-catalytic systems have attracted considerable interest owing to their high catalytic activity, low toxicity, and mild reaction conditions. Herein, we reported the synthesis of a bifunctional TEMPO-IsoNTA organocatalyst featuring a pyridyl group as an anchoring site and a TEMPO radical as a catalytic active site. By using the topologically isomorphic structures of MIL-101(Fe) and MIL-101(Cr) as co-catalysts, these MOF/TEMPO-IsoNTA systems enable the efficient aerobic oxidation of various alcohols to their corresponding aldehydes or ketones under mild conditions. Notably, the MIL-101(Fe)/TEMPO-IsoNTA system exhibits superior catalytic activity, thanks to their redox-active Fe^III^-oxo nodes, which facilitate the regeneration of TEMPO-IsoNTA. Our research not only solves the problem of potential heavy metal contamination in the TEMPO-based homogeneous catalytic system, but also enriches the understanding of synergism of MOFs/organocatalysts.

## 1. Introduction

Metal–organic frameworks (MOFs) are envisaged as opportunistic materials for heterogeneous catalysis due to to their controllable pore size, high surface area, and diverse synthetic strategy [1,2]. The porosity of MOFs yields a large internal surface area, greatly enhancing the catalytic activity of substrates [3]. Uniform internal pores in MOFs constrain the passing through of reactants to improve the reaction selectivity. Particularly, the metal nodes of MOFs serving either as potential catalytic sites or anchoring sites for functional molecules could play crucial roles in catalysis [4,5]. Despite the tremendous effort devoted to developing various MOF-based catalysts, conceiving the sophisticated methods to build up MOF/organocatalyst co-catalytic systems with desirable stability, selectivity and solvent tolerance is far away from realistic [6,7,8,9,10].

Among the numerous MOFs, the MIL-101 family (MIL, denoted as Material of Institute Lavoisier) is particularly interesting owing to its excellent chemical/thermal stabilities, large pore volume, and extremely high surface area [11,12]. For example, MIL-101(Fe/Cr) is constructed by Fe^III^/Cr^III^-oxo clusters with crosslinker terephthalate (BDC) linkers to give birth to micropores and mesopores with an aperture of 0.86 nm and 2.9 nm, respectively. Thehybrid micro/mesopore structures of MIL-101(Fe/Cr), together with their merits of a high density of unsaturated metal nodes, redox capacity, and elemental abundance, make them promising heterogenous catalysts [12]. For instance, the catalytic properties of MIL-101(Fe/Cr) have been assessed for hydrocarbon oxidation [13], isomerization of dicyclopentadiene [14], allylic oxidation of alkenes [15], epoxidation of styrene [16], etc. It is well-established that MIL-101(Fe) and MIL-101(Cr) possess dissimilar catalytic activity, despite the topologically isomorphic structure. The underlying reasons for the different catalytic activity mainly lie in the distinct redox ability, coordination-unsaturated sites, and Lewis acidity of Fe^III^-oxo and Cr^III^-oxo metal nodes. Concretely, the Fe^III^-oxo nodes in MIL-101(Fe) can undergo Fe^3+^/Fe^2+^ redox shuttle without changing the crystal structure of MOFs, thus boosting its redox catalytic activity, while the unsaturated and Lewis acidic Cr^III^-oxo nodes in MIL-101(Cr) render the catalytic activity in many chemical reactions [17,18].

The selective oxidation of alcohols to their corresponding aldehydes or ketones is an important reaction in the synthesis of fine chemicals and organic intermediates [19]. Despite the great achievements in this field, there is an urgent need for the development of efficient, highly selective, and reusable catalytic systems to meet the criteria of green chemistry [20,21,22]. Aerobic oxidation with molecular oxygen as the terminal oxidant has attracted great attention because of its abundance, low cost, environmental friendliness, and sustainability [21]. A family of stable nitroxide radicals, TEMPO ((2,2,6,6-tetramethylpiperidine-1-yl)oxyl) and its derivatives, represents versatile organocatalysts for aerobic oxidation of alcohols by reason of the high catalytic activity, excellent selectivity and mild reaction conditions [23,24,25,26]. For example, Lagerblom et al. [27] established a Fe(NO_3_)_3_/2,2′-bipyridine/TEMPO co-catalytic system, achieving selective aerobic oxidation of aliphatic, benzylic primary alcohols to aldehydes or carboxylic acids. Yin et al. [28] developed a 4-acetamido-TEMPO/FeCl_3_/NaNO_2_ co-catalytic system, enabling the oxidation of a range of benzyl alcohols, heteroatomic alcohols and aliphatic alcohols with oxygen under mild conditions. Stahl et al. [29] systematically explored the catalytic performances of a TEMPO/Cu system for the oxidation of aromatic alcohols and demonstrated that the catalytic activity of the TEMPO/Cu system highly relies on the copper salts, ligands, and bases. Note that transition metal salts are often required in many TEMPO-based homogeneous catalytic systems, suffering from difficulties in catalyst recovery and heavy metal contamination. One way to address this problem is the use of MOFs as co-catalysts, and several MOF/TEMPO catalytic systems have been reported [30,31,32,33]. Despite the great achievements made, insights into the catalytic active sites, the role of metal nodes, as well as the synergistic effect remain largely unexplored.

In this work, we designed and synthesized a pyridyl-functionalized TEMPO radical (TEMPO-IsoNTA, N-(2,2,6,6-tetramethylpiperidine-1-yloxyl)isonicotinamide) by an amidation reaction between isonicotinic acid (IsoNTA) and 4-amino-TEMPO, and used MIL-101(Fe) or MIL-101(Cr) as co-catalysts for the selective aerobic oxidation of various alcohols. Compared with TEMPO-IsoNTA, the MIL-101(Fe)/TEMPO-IsoNTA co-catalyst shows enhanced catalytic activity. In contrast, the use of MIL-101(Cr) as a co-catalyst significantly deceases the catalytic performance for alcohol transformation. Given the topologically isomorphic structures of MIL-101(Fe) and MIL-101(Cr) (Figure 1), it is implied that the higher catalytic activity of MIL-101(Fe)/TEMPO-IsoNTA compared to that of MIL-101(Cr)/TEMPO-IsoNTA is mainly attributed to the Fe^3+^/Fe^2+^ redox couple, which facilitates the regeneration of TEMPO-IsoNTA, thus boosting the catalytic activity. Our findings have established a valuable understanding of TEMPO/MOF catalytic systems for the effective oxidation of alcohols, to increase the reactivity, simplify the synthetic routes and extend the substrate scope.

## 2. Results and Discussion

### 2.1. Synthesis and Characterization of TEMPO-IsoNTA, MIL-101(Fe) and MIL-101(Cr)

TEMPO-IsoNTA was synthesized by an amidation reaction between isonicotinic acid and 4-amino-TEMPO with a yield of 75% (see Appendix A for details). The successful synthesis of the TEMPO-IsoNTA ligand was confirmed by ^1^HNMR, ^13^CNMR, and MS spectra (Appendix A).

MIL-101(Fe) and MIL-101(Cr) were prepared by the solvothermal method. As shown in the SEM image in Figure 2a, MIL-101(Fe) displays octahedral microparticles with an average size of 2 µm. On closer inspection, we observed that the (111) crystallographic plane looks somewhat concave, whereas the MIL-101(Cr) displays a uniform octahedral shape with a smaller size (ca. 500 nm), as depicted in the SEM image in Figure 2b. The crystallinity of MIL-101(Fe) and MIL-101(Cr) was confirmed by powder XRD spectra (Figure 2c). The diffraction patterns of MIL-101(Fe) and MIL-101(Cr) are very similar, and match well with that of the simulated pattern from the single crystal X-ray data, indicating the high phase purity of the as-synthesized MIL-101(Fe) and MIL-101(Cr). To evaluate the assessable inner surfaces of MIL-101(Fe) and MIL-101(Cr), nitrogen adsorption/desorption isotherms were recorded at 77 K. As illustrated in Figure 2d, the results reveal that the samples are compatible with type I isotherms, with a BET (Brunauer–Emmett–Teller) surface area of 1570 m^2^/g and 2928 m^2^/g for MIL-101(Fe) and MIL-101(Cr), respectively. The higher value of the BET surface area of MIL-101(Cr) compared to that of MIL-101(Fe) might be attributed to the excellent thermal stability and full activation of MIL-101(Cr). Moreover, the pore size distribution profiles calculated by nonlocal density functional theory (NL-DFT) indicated that both MIL-101 MOFs comprise micropores and mesopores (Appendix A).

### 2.2. Adsorption Properties of TEMPO-IsoNTA with MIL-101(Fe) or MIL-101(Cr)

As stated before, the pyridine group in TEMPO-IsoNTA is expected to coordinate to the open metal sites in MOFs, which are µ_3_-OFe^III^_3_ and µ_3_-OCr^III^_3_ in MIL-101(Fe) and MIL-101(Cr), respectively. To this end, we monitored the UV–Vis spectra of TEMPO-IsoNTA before and after the addition of MOFs. As depicted in Figure 3a, the absorption band of TEMPO-IsoNTA solution in the visible region dramatically declined after the removal of MIL-101(Fe), indicative of a strong adsorption of TEMPO-IsoNTA with MIL-101(Fe). The amount of adsorbed TEMPO-IsoNTA calculated by a standard working curve is 7.21 × 10^−4^ mmol per mg (Appendix A). In the case of MIL-101(Cr) as the adsorbents, almost all the TEMPO-IsoNTA molecules (1.33 × 10^−3^ mmol per mg) were trapped in the MOFs, as indicated by the absence of an absorption band in the UV–Vis spectra. The better adsorption capacity of MIL-101(Cr) is clearly visualized by the transparent filtered solution, whereas a light-yellow color was observed for a filtered solution by removing MIL-101(Fe). The trapped TEMPO-IsoNTA molecules in MOFs were further confirmed by FT-IR spectra. Detailed comparisons of the FT-IR spectra show a weak C–H vibration signal around 3000 cm^−1^ (marked by the yellow column), which arose from TEMPO-IsoNTA, indicating the successful incorporation of TEMPO-IsoNTA into MIL-101(Fe). This C–H vibration signal becomes more pronounced when using MIL-101(Cr) as adsorbents. Moreover, the peak at 1540 cm^−1^ (marked by the green column), which can be assigned to the C=O of amide, appeared in the spectrum of MIL-101(Cr)/TEMPO-IsoNTA, indicative of the encapsulation of TEMPO-IsoNTA. The superior adsorption capacity of MIL-101(Cr) for TEMPO-IsoNTA is in line with the UV–Vis spectra results. It is worth noting that the vanished N–H stretch bands (3300 cm^−1^) in both MIL-101 adsorbents hint that, beside the pyridyl–metal bond, TEMPO-IsoNTA might also coordinate to µ_3_-OFe^III^_3_ and µ_3_-OCr^III^_3_ nodes by a metal–amide bond.

### 2.3. Synergistic Catalytic Studies of MIL-101(Fe) and MIL-101(Cr)

Transition-metal-based TEMPO catalyst is one of the most efficient catalytic systems toward the oxidation of various alcohols, exhibiting excellent selectivity. However, the homogeneous catalytic systems have problems regarding the recovery and reuse of metal salts, as well as the contamination of organic products by trace amounts of heavy metals. The highly accessible metal nodes, as well as the large pores of MIL-101 MOFs, encouraged us to study their catalytic activity toward the oxidation of alcohols. To this end, we chose the MIL-101(Fe or Cr)/TEMPO-IsoNTA co-catalytic system assisted by TBN (tert-butyl nitrite) as NO sources to active oxygen molecules, and examined their catalytic activity by the selective oxidation of benzyl alcohol to benzaldehyde as a model reaction; the results are summarized in Table 1. Initially, the reaction was carried out in a closed vial with 1.5 mL of benzotrifluoride (PhCF_3_), benzyl alcohol (0.3 mmol), TEMPO-IsoNTA as a homogeneous catalyst (5 mol%), and TBN (14 mol%) under oxygen atmosphere (balloon) at 80 °C for 1.5 h, and only 23% of benzyl alcohol was converted to benzaldehyde (Entry 1). Under identical conditions, the addition of MIL-101(Fe) as co-catalysts (5 mol%) markedly increased the catalytic activity of TEMPO-IsoNTA, and full conversion of benzyl alcohol was achieved with excellent selectivity (Entry 2). Surprisingly, the addition of MIL-101(Cr) did not increase the catalytic activity of TEMPO-IsoNTA, but a much lower conversion (9%, Entry 3) was observed. At first glance, this result is inconsistent with results of the higher encapsulation of TEMPO-IsoNTA in MIL-101(Cr) compared to that of MIL-101(Fe). A possible reason might be that the strong Lewis acidity of Cr(III) allows the coordination of TEMPO-IsoNTA by N-O^•^→Cr(III), leading to a decreased concentration of freely available N-O^•^ radical, thus lowering the catalytic activity of MIL-101(Cr)/TEMPO-IsoNTA. Additional experimental and theoretic work are needed to fully understand this unexpected effect.

It has been well-established that the NO released from TBN was oxidized by oxygen into NO_2_, which is able to oxidize TEMPO radicals into TEMPO oxoammonium [23]. The TEMPO oxoammonium is a key intermediator for the oxidation of alcohols. By increasing the amount of TBN from 14 mol% to 25 mol%, the conversion of benzyl alcohol, catalyzed by either TEMPO-IsoNTA or MIL-101(Cr)/TEMPO-IsoNTA, was indeed largely increased (Entry 1a, 3a). This finding implies that the catalytic activity of MIL-101/TEMPO-IsoNTA systems relies not only on trapped TEMPO-IsoNTA in MOFs, but also on the metal nodes. For example, the redox capability of Fe^3+^/Fe^2+^ can be a key factor for the enhanced catalytic activity of MIL-101(Fe), which will be discussed in the section regarding the catalytic mechanism. Controlled experiments (Entries 4–9) suggested that MIL-101(Fe), TEMPO-IsoNTA, and TBN initiator are all required to achieve a high catalytic performance toward the aerobic oxidation of benzyl alcohol. According to He’s work [23], TBN can generate NO_2_ and NO under thermal conditions, which facilitates the conversion of the Fe^3+^/Fe^2+^ redox couple, thus promoting the catalytic efficiency of the MIL-101(Fe)/TEMPO-IsoNTA system. Moreover, by using the commercially available TEMPO radical as a catalyst, a benzyl alcohol conversion of only 42% was achieved, indicating the importance of the anchoring site of TEMPO-IsoNTA for superior catalytic activity.

To gain more information about the catalytic kinetics of MIL-101/TEMPO-IsoNTA, plots of conversion versus time were recorded and are shown in Figure 4a. At the initial stage of the reaction (e.g., 0–30 min), the conversion of benzyl alcohol was almost the same for MIL-101(Fe)/TEMPO-IsoNTA and MIL-101(Cr)/TEMPO-IsoNTA, that is, 17% and 16% after 30 min, respectively. The comparable catalytic efficiency at this stage is likely attributed to the homogeneous TEMPO-IsoNTA/TBN catalytic system. With the prolongation of the reaction time, the catalytic efficiency of the MIL-101(Fe)/TEMPO-IsoNTA system was significantly higher than that of MIL-101(Cr)/TEMPO-IsoNTA, indicating that the synergetic effect stemming from the partially anchored TEMPO-IsoNTA on the Fe^III^-oxo node was mainly responsible for the enhanced catalytic activity. The role of Fe^3+^ for TEMPO-mediated aerobic oxidation has been reported in several homogeneous catalytic systems [27,34,35], where the redox Fe^3+^/Fe^2+^ couple is believed to accelerate the recovery of TEMPO from TEMPOH by H-abstraction. The catalytic mechanism of the redox-active metal node promoting the aerobic oxidation of alcohols will be discussed later.

To evaluate the stability and recyclability of MIL-101, three runs of benzyl alcohol oxidation were carried out with MIL-101(Fe)/TEMPO-IsoNTA and MIL-101(Cr)/TEMPO-IsoNTA under optimal reaction conditions. Between each run, MIL-101 was removed from the reaction solution by centrifugation, washed with benzotrifluoride, and directly used for next run. Full conversion of benzyl alcohol was achieved in the second run for MIL-101(Fe)/TEMPO-IsoNTA, while the conversion dropped to 60% in the third run. To understand the decreased catalytic activity of MIL-101(Fe), the XRD pattern (Appendix A) and high-resolution Fe 2p XPS spectrum (Appendix A) of fresh and used MIL-101(Fe) were recorded. After three runs, some of the diffraction peaks (see Appendix A, marked by blue stars) of MIL-101(Fe) became less pronounced compared to fresh MIL-101(Fe), indicating the decreased crystallinity of used MIL-101(Fe). In the XPS spectra of fresh MIL-101(Fe), Fe 2p_3/2_ and Fe 2p_1/2_ peaks associated with their satellite peaks (718.2 and 731.1 eV) were observed. The two sets of deconvoluted peaks located at 709.6 and 722.8 eV and at 711.1 and 724.8 eV can be assigned to Fe^2+^ and Fe^3+^, respectively [36]. Similar deconvoluted peaks can be also obtained in the spectrum of used MIL-101(Fe), indicating the presence of Fe^3+^ and Fe^2+^. Detailed analysis of the mean relative areas suggested that the ratio of Fe^3+^/Fe^2+^ increased from 1:1 to 1.5:1 after use. The decreased crystallinity and imbalanced Fe^3+^/Fe^2+^ might be responsible for the decreased catalytic activity of MIL-101(Fe).

We further expanded the substrate scope to a variety of alcohols with different dimensions. As shown in Table 2, MIL-101(Fe)/TEMPO-IsoNTA exhibited superior catalytic activity and selectivity toward the oxidation of para-substituted benzyl alcohols, regardless of electron-withdrawing or electron-donating substitutions. MIL-101(Fe)/TEMPO-IsoNTA also enables the oxidation of secondary aromatic alcohols to their corresponding ketones with excellent catalytic efficiency (Table 2, Entries 4–5). For S/O-containing heteroatomic alcohols, as well as secondary aliphatic alcohols, full conversion can be achieved with slightly longer reaction times (Table 2, Entries 6–8). Similar to the previous results, for all of the tested alcohols, MIL-101(Fe)/TEMPO-IsoNTA showed a higher catalytic activity than that of MIL-101(Cr)/TEMPO-IsoNTA.

### 2.4. Catalytic Mechanism: Redox-Active Metal–Organic Framework Nodes Boost the Catalytic Activity

Considering the topologically isomorphic structures of MIL-101(Fe) and MIL-101(Cr), the superior catalytic activity of MIL-101(Fe)/TEMPO-IsoNTA compared to that of MIL-101(Cr)/TEMPO-IsoNTA is mainly attributed to redox-active Fe^III^-oxo nodes. Together with previously reported TEMPO/TBN catalytic cycles [23], a plausible catalytic mechanism based on Fe^3+^/Fe^2+^ redox couple boosted catalytic activity is proposed in Figure 5. The functionalized TEMPO-IsoNTA not only serves as catalytic active sites (TEMPO radical), but also offers anchoring sites (pyridyl group) to coordinate to MIL-101(Fe). The rich Fe^III^-oxo nodes of MIL-101(Fe) are partially anchored by TEMPO-IsoNTA, leading to freely available Fe^III^-oxo nodes for the synergistic interaction with TEMPO radicals. The thermally unstable TBN releases NO and further oxide to NO_2_ by molecular oxygen. Either the anchored or free TEMPO-IsoNTA (2) is readily activated and oxidized by NO_2_ to form the TEMPO oxoammonium species (3), which is a reactive intermediator for H-abstraction from alcohols to give target aldehydes (or ketones). Meanwhile, the TEMPO oxoammonium species is converted to its reduced hydroxylamine species state, TEMPOH-IsoNTA (1). TEMPOH-IsoNTA can be oxidized to TEMPO-IsoNTA by Fe^III^-oxo nodes, thus completing the catalytic cycle. The regeneration of Fe^III^-oxo nodes is achieved through the following reaction [35]:Fe^2+^ + NO_2_ + 2H^+^→Fe^3+^ + NO + H_2_O. 

Clearly, the redox-active Fe^III^-oxo nodes in MIL-101(Fe) accelerate the regeneration of TEMPO-IsoNTA from the reduced state TEMPOH-IsoNTA, thus boosting the catalytic efficiency of the aerobic oxidation of alcohols.

## 3. Materials and Methods

### 3.1. Synthesis of TEMPO-IsoNTA

TEMPO-IsoNTA was synthesized based on a previously reported method with slight modification [37]. To a solution of isonicotinic acid (369.3 mg, 3 mmol) in anhydrous CH_2_Cl_2_ (15 ml) at 0 °C, DCC (742.2 mg, 3.6 mmol) and HOBt (486.5 mg, 3.6 mmol) were added and stirred for 30 min under nitrogen atmosphere. Then, 4-amino-TEMPO (513.8 mg, 3.0 mmol) in anhydrous CH_2_Cl_2_ (7.5 mL) was added dropwise to the above solution. The mixture was warmed to room temperature and further stirred for 12 h under nitrogen atmosphere. The reaction mixture was filtered through a pad of Celite and washed with CH_2_Cl_2_. The solution was evaporated to obtain the crude product as a thick oil, which was purified on a silica gel column (CH_2_Cl_2_:MeOH = 50:1) to collect the pure product as an orange solid (623 mg, 75%). Because of the paramagnetic nature of the nitroxide radical, conventional NMR spectroscopy cannot offer valuable structural information. Therefore, phenylhydrazine was used to reduce the nitroxide radial to N-hydroxy amine before NMR measurements. In general, to a solution of TEMPO-IsoNTA (5 mg) in DMSO_d6_ (0.5 mL) in an NMR tube, 5 μL of phenylhydrazine was added. After 30 min, the NMR spectrum of the corresponding N-hyroxyl amine was collected, and the purity of TEMPO-IsoNTA, determined by NMR, was higher than 97%.

### 3.2. Synthesis of MOFs

#### 3.2.1. Synthesis of MIL-101(Fe)

MIL-101(Fe) was synthesized according to a published procedure [38]. First, 166.13 mg of linker BDC (1 mmol) and 675 mg of FeCl_3_·6H_2_O (2.5 mmol) were added to 15 mL of DMF. The suspension was ultra-sonicated until the solid was completely dissolved. Then, the solution was transferred into an autoclave and heated at 110 °C for 24 h. After cooling to room temperature, the synthesized reddish-brown MIL-101(Fe) was separated from the reaction mixture by centrifugation and washed thoroughly with DMF, H_2_O and EtOH to remove any unreacted raw materials within the pores. Subsequently, the obtained solid was soaked in EtOH for 24 h in a soxhlet extractor, and finally dried under vacuum at 60 °C.

#### 3.2.2. Synthesis of the MIL-101(Cr)

MIL-101(Cr) was synthesized following a previously described protocol with some modifications [39]. Typically, 400 mg of Cr(NO_3_)_3_·9H_2_O (1.0 mmol) and 166.1 mg of H_2_BDC (1.0 mmol) were added to a Teflon-lined autoclave containing 7.2 mL of H_2_O and 44 µL of HF. The suspension was ultra-sonicated until the solid was completely dispersed. Then, the autoclave was heated at 210 °C for 24 h. When the reaction was completed and slowly cooled to room temperature, a green powder was obtained. The resulting precipitate was collected from the reaction mixture by centrifugation and washed thoroughly with DMF, H_2_O and EtOH to remove any unreacted organic species within the pores. After that, the obtained solid was further purified by solvothermal treatment in EtOH. Finally, the resulting solid was dried under vacuum at 60 °C.

### 3.3. Studies of the Adsorption Properties of MIL-101(Fe) and MIL-101(Cr) for TEMPO-IsoNTA

To a 10 mL screw-capped vial, MIL-101(Fe) (10.7 mg, 5 mol%) or MIL-101(Cr) (10.8 mg, 5 mol%)was added, followed by 5 mL of benzotrifluoride solution of TEMPO-IsoNTA (10 mM) under sonication until the solid was completely dispersed. Then, the vial was sealed and placed in an oven (80 °C) for 1.5 h. After cooling to room temperature, the solid was removed by centrifugation, the clear solution was analyzed by an ultraviolet–visible spectrophotometer to test the adsorption properties of MIL-101(Fe) and MIL-101(Cr) for TEMPO-IsoNTA.

### 3.4. Studies of the Synergistic Catalytic Properties of MIL-101(Fe)/TEMPO-IsoNTA and MIL-101(Cr)/TEMPO-IsoNTA

Typical procedure for the aerobic oxidation of alcohols: To a 10 mL screw-capped vial, benzyl alcohol (0.3 mmol), MIL-101(Fe)/MIL-101(Cr) (0.015 mmol), TEMPO-IsoNTA (0.015 mmol), PhCF_3_ (1.5 mL, saturated with O_2_), and TBN (0.06 mmol) were added, then the vial was sealed and placed in an oven (80 °C) for the desired time. After cooling to room temperature, p-nitrobenzene (internal standard, 0.3 mmol) was added, and the catalyst was removed by centrifugation. The clear solution was analyzed by GC without further purification. Regarding the recycling experiments, benzyl alcohol was used as the substrate. The catalysts were collected by centrifugation, washed with ethanol twice and dried under vacuum.

## 4. Conclusions

In summary, we have synthesized a bifunctional TEMPO-IsoNTA organocatalyst, where the pyridyl group serves as an anchoring site, and the TEMPO moiety serves as a catalytic active site. By using the topologically isomorphic structures of MIL-101(Fe) and MIL-101(Cr) as co-catalysts, we have investigated the catalytic activity of TEMPO-IsoNTA for the selective aerobic oxidation of various alcohols under mild conditions. The MIL-101(Fe)/TEMPO-IsoNTA co-catalytic system exhibits a higher catalytic activity than that of the MIL-101(Cr)/TEMPO-IsoNTA system. Detailed experimental studies suggested that the redox-active Fe^III^-oxo nodes in MIL-101(Fe) facilitate the regeneration of TEMPO-IsoNTA from the reduced state TEMPOH-IsoNTA, thus boosting the catalytic activity. This co-catalytic system not only solves the problem of potential heavy metal contamination in the TEMPO-based homogeneous catalytic system, but also realizes the transition from homogeneous catalysis to heterogeneous catalysis. Our findings provide new ideas for the design and synthesis of ligand-based organic molecule/MOF co-catalytic system.

## Figures and Tables

**Figure 1 molecules-28-00593-f001:**
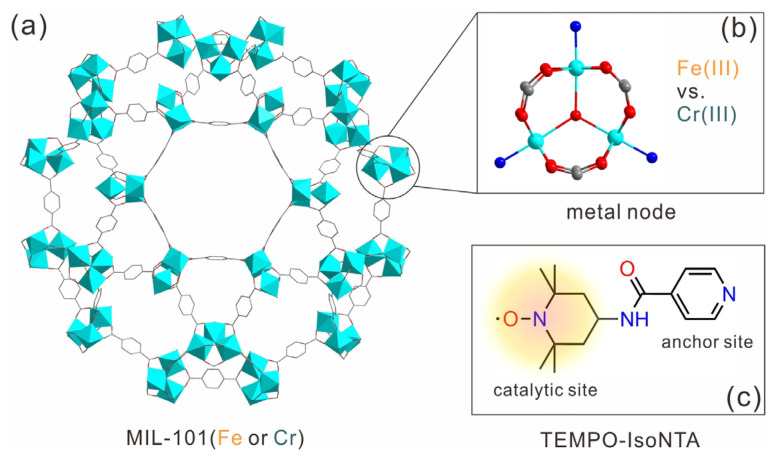
(**a**) Crystal structure of MIL-101(Fe/Cr); (**b**) open metal sites in MIL-101(Fe/Cr); (**c**) molecular structure of the TEMPO-IsoNTA bearing catalytic site and anchoring site. C: grey; O: red; Fe^3+^ or Cr^3+^: cyan; N: blue.

**Figure 2 molecules-28-00593-f002:**
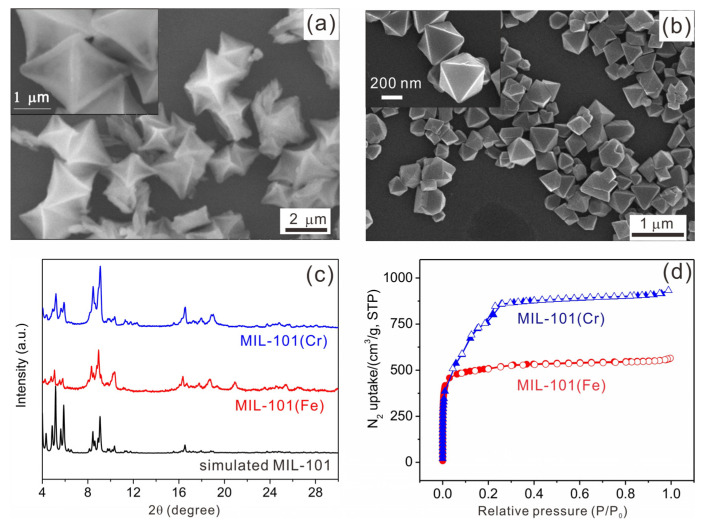
SEM images of (**a**) MIL-101(Fe) and (**b**) MIL-101(Cr); (**c**) powder XRD patterns of MIL-101(Cr), MIL-101(Fe) and simulated XRD pattern from MIL-101; (**d**) N_2_ adsorption–desorption isotherms of MIL-101(Fe) (red curve) and MIL-101(Cr) (blue curve).

**Figure 3 molecules-28-00593-f003:**
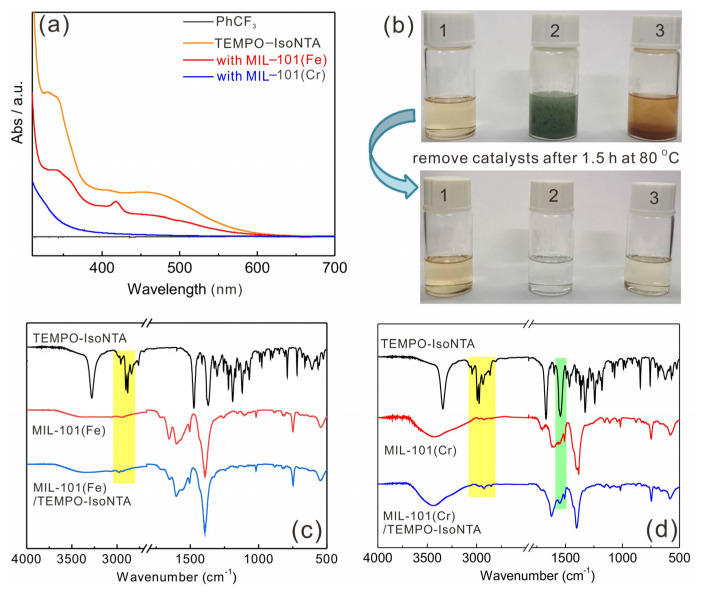
(**a**) UV–Vis adsorption spectra of TEMPO-IsoNTA with MIL-101(Fe) or MIL-101(Cr); (**b**) photographs of the color changes of TEMPO-IsoNTA solution (1), before (up) and after (down) adsorption with MIL-101(Cr) (2) and MIL-101(Fe) (3); (**c**) FT-IR spectra of TEMPO-IsoNTA, MIL-101(Fe) and MIL-101(Fe)/TEMPO-IsoNTA; (**d**) FT-IR spectra of TEMPO-IsoNTA, MIL-101(Cr) and MIL-101(Cr)/TEMPO-IsoNTA.

**Figure 4 molecules-28-00593-f004:**
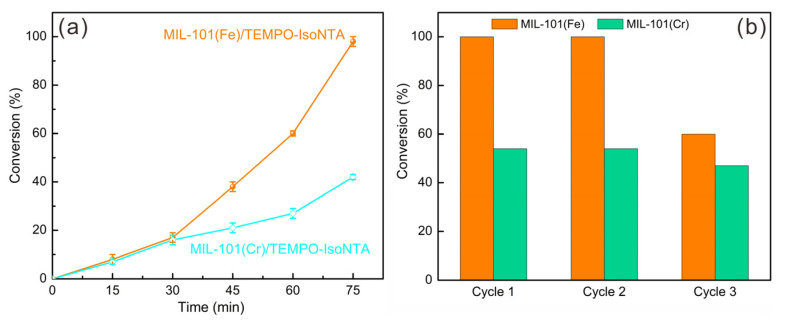
(**a**) Kinetic curves of benzyl alcohol to benzaldehyde by MIL-101(Fe)/TEMPO-IsoNTA (orange line) and MIL-101(Cr)/TEMPO-IsoNTA (light blue line) under identical reaction conditions; (**b**) catalytic conversion of benzyl alcohol catalyzed by MIL-101(Fe) (orange column) and MIL-101(Cr) (green column) during three runs.

**Figure 5 molecules-28-00593-f005:**
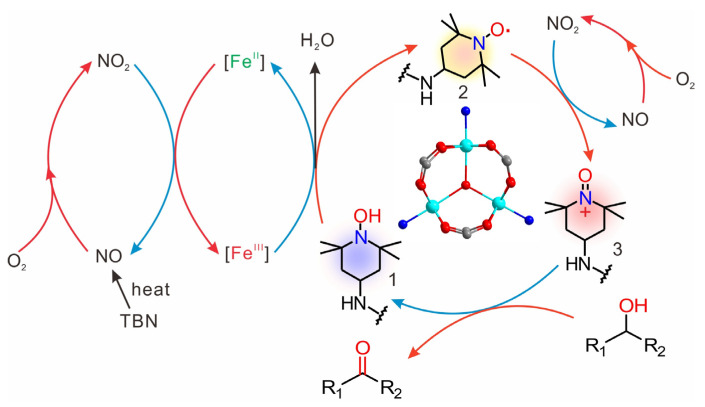
A plausible catalytic mechanism for the aerobic oxidation of alcohols by MIL-101(Fe)/TEMPO-IsoNTA.

**Table 1 molecules-28-00593-t001:** Aerobic oxidation of benzyl alcohol under various conditions.

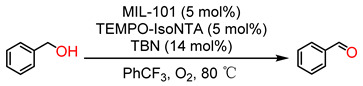						
Entry	Catalyst	Co-Catalyst	Activator	Time (h)	Con. (%)	Sel. (%)
1	—	TEMPO-IsoNTA	TBN	1.5	23/60 ^a^	>99
2	MIL-101(Fe)	TEMPO-IsoNTA	TBN	1.5	100/100 ^a^	>99
3	MIL-101(Cr)	TEMPO-IsoNTA	TBN	1.5	9/51 ^a^	>99
4	MIL-101(Fe)	—	TBN	1.5	5	>99
5	MIL-101(Cr)	—	TBN	1.5	5	>99
6	MIL-101(Fe)	TEMPO-IsoNTA	—	1.5	4	>99
7	MIL-101(Cr)	TEMPO-IsoNTA	—	1.5	4	>99
8	MIL-101(Fe)	—	—	1.5	<1	—
9	MIL-101(Cr)	—	—	1.5	<1	—
10	MIL-101(Fe)	TEMPO	TBN	1.5	42	>99

Reaction conditions: benzyl alcohol (0.3 mmol), catalyst (5 mol%), co-catalyst (5 mol%), TBN (14 mol%), PhCF_3_ (1.5 mL); reaction time: 1.5 h. Conversion and selectivity were determined by GC analysis with p-nitrobenzene as an internal standard. ^a^: TBN (25 mol%).

**Table 2 molecules-28-00593-t002:** MIL-101(Fe) or MIL-101(Cr) catalyzed aerobic oxidation of various alcohols.

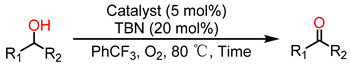
Entry	Product	Time (h)	Fe-MOF Cat.Con. (%)	Cr-MOF Cat.Con. (%)	Sel. (%)
1	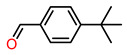	1.5	>99	8	>99
2	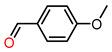	1.5	>99	30	>99
3	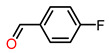	1.5	>99	25	>99
4	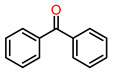	1.5	>99	70	>99
5	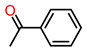	1.5	>99	7	>99
6	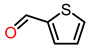	4	>99	70	>99
7	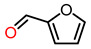	4	>99	60	>99
8	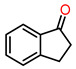	6	>99	18	>99

Reaction conditions: benzyl alcohol (0.3 mmol), catalyst (5 mol%), TBN (20 mol%), PhCF_3_ (1.5 mL). Conversion and selectivity were determined by GC analysis with p-nitrobenzene as an internal standard. Fe-MOF Cat. represents MIL-101(Fe)/TEMPO-IsoNTA; Cr-MOF Cat. represents MIL-101(Cr)/TEMPO-IsoNTA.

## Data Availability

The data presented in this study are available in article and Appendix A.

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
