# Peer review of "Enhanced Catalytic Activity of TEMPO-Mediated Aerobic Oxidation of Alcohols via Redox-Active Metal–Organic Framework Nodes"

_molecules, 2023, doi:10.3390/molecules28020593_

Round 1

Reviewer 1 Report

The present manuscript reports on the “Redox-Active Metal-Oganic Framework Nodes Boost Catalytic Activity of TEMPO-Mediated Aerobic Oxidation of Alcohols”. The work is of some interest but seems to be too primitive and lacks in proper scientific support and justification. There are many reports for catalysis. Thus, in my opinion, the manuscript in its present form cannot be considered for publication. I recommend major revision.

Following are some of the comments/suggestions which will be useful to the authors.

1. write the oxidation state of transition metal ions like Cr+3 or Cr(III).

2. The references are not correct. For example “Lagerblom et al. established a Fe(NO3)3/2,2’-bipyri-66 dine/TEMPO co-catalytic system, achieving selective aerobic oxidation of aliphatic, benzylic primary alcohols to aldehydes or carboxylic acids [27]. Yin et al. developed a 4-acetamido-TEMPO/FeCl3/NaNO2 co-catalytic system, enabling the oxidation of a range of benzyl alcohols, heteroatomic alcohols and aliphatic alcohols with oxygen under mild conditions [28]. Stahl et al. systematically explored the catalytic performances of a TEMPO/Cu system for oxidation of aromatic alcohols and demonstrated that the catalytic activity of TEMPO/Cu system highly relies on the copper salts, ligands, bases [29]” but the references are

27. Agerblom, K.; Wrigstedt, P.; Keskivali, J.; Parviainen, A.; Repo, T., Iron-Catalysed Selective Aerobic Oxidation of Alcohols to Carbonyl and Carboxylic Compounds. Chempluschem 2016, 81, 1160-1165.

28. Yin, W.; Chu, C.; Lu, Q.; Tao, J.; Liang, X.; Liu, R., Iron Chloride/4-Acetamido-TEMPO/Sodium Nitrite-Catalyzed Aerobic Oxi-dation of Primary Alcohols to the Aldehydes. Advanced Synthesis & Catalysis 2010, 352, 113-118.

29. Hoover, J. M.; Ryland, B. L.; Stahl, S. S., Copper/TEMPO-Catalyzed Aerobic Alcohol Oxidation: Mechanistic Assessment of Different Catalyst Systems. ACS Catalysis 2013, 3, 2599-2605.

Correct all references from the whole manuscript according to the suitability.

3. Write the reference numbers after “et al” in the text. For example “Lagerblom et al. [27] established a Fe(NO3)3/2,2’-bipyri-66 dine/TEMPO co-catalytic system, achieving selective aerobic oxidation of aliphatic, benzylic primary alcohols to aldehydes or carboxylic acids”.

4. write the complete name in the first appearance in the manuscript. The authors did follow this pattern. For example, in line 16 and 18.

5. Units are not correct in Figure 2, 3, and 4. Write it as for example “wavelength (nm)”.

6.Correct all units. The units are not correct such as some where the unit of time is used mins and somewhere minutes.

7. The authors did not represent the 2θ values in Figure 2(c).

8. How can you confirm that the product is aldehydic. It may be possible that the alcoholic group was converted into carboxylic groups.

9. Title of article should be improved.

10. It is very important to write the synthetic scheme which is missing in this manuscript.

11. Correct the molecular formulas. Some formulas are incorrect such as CrCl3•6H2O, FeCl3•6H2O

12. There are many previous works published for catalysis. The authors seem deliberately avoid those papers. The authors need to acknowledge the previous literature and compare their work with the similar ones and other systems in the literature and demonstrate their research outcomes in terms of advantages and disadvantages. Some of studies are given below need to cited;

https://doi.org/10.1016/j.cplett.2020.137645;

https://doi.org/10.1016/j.inoche.2022.109870;

https://doi.org/10.1039/D1RA09380K

Author Response

Comments and Replies

The reviewer's questions and comments (italic font), our replies (normal front), followed by the changes made in the text (green font).

Reviewer 1:

The present manuscript reports on the “Redox-Active Metal-Oganic Framework Nodes Boost Catalytic Activity of TEMPO-Mediated Aerobic Oxidation of Alcohols”. The work is of some interest but seems to be too primitive and lacks in proper scientific support and justification. There are many reports for catalysis. Thus, in my opinion, the manuscript in its present form cannot be considered for publication. I recommend major revision.

Following are some of the comments/suggestions which will be useful to the authors.

1write the oxidation state of transition metal ions like Cr+3 or Cr(III).

Reply: Thanks for your kind reminder. We have corrected the oxidation state of transition metal ions, and Fe(III) and Cr(III) have been used in the revised manuscript.

  1. The references are not correct. For example “Lagerblom et al. established a Fe(NO3)3/2,2’-bipyri-66 dine/TEMPO co-catalytic system, achieving selective aerobic oxidation of aliphatic, benzylic primary alcohols to aldehydes or carboxylic acids [27]. Yin et al. developed a 4-acetamido-TEMPO/FeCl3/NaNO2 co-catalytic system, enabling the oxidation of a range of benzyl alcohols, heteroatomic alcohols and aliphatic alcohols with oxygen under mild conditions [28]. Stahl et al. systematically explored the catalytic performances of a TEMPO/Cu system for oxidation of aromatic alcohols and demonstrated that the catalytic activity of TEMPO/Cu system highly relies on the copper salts, ligands, bases [29]” but the references are
  2. Agerblom, K.; Wrigstedt, P.; Keskivali, J.; Parviainen, A.; Repo, T., Iron-Catalysed Selective Aerobic Oxidation of Alcohols to Carbonyl and Carboxylic Compounds. Chempluschem 2016, 81, 1160-1165.
  3. Yin, W.; Chu, C.; Lu, Q.; Tao, J.; Liang, X.; Liu, R., Iron Chloride/4-Acetamido-TEMPO/Sodium Nitrite-Catalyzed Aerobic Oxi-dation of Primary Alcohols to the Aldehydes. Advanced Synthesis & Catalysis 2010, 352, 113-118.
  4. Hoover, J. M.; Ryland, B. L.; Stahl, S. S., Copper/TEMPO-Catalyzed Aerobic Alcohol Oxidation: Mechanistic Assessment of Different Catalyst Systems. ACS Catalysis 2013, 3, 2599-2605.

Correct all references from the whole manuscript according to the suitability.

Reply: Thank you for pointing out our mistakes. After careful check, we found the name of first author in Ref. 27 was indeed wrong and has been revised. The other mentioned numberings of the Refs are all correct.

  1. Write the reference numbers after “et al” in the text. For example “Lagerblom et al. [27] established a Fe(NO3)3/2,2’-bipyri-66 dine/TEMPO co-catalytic system, achieving selective aerobic oxidation of aliphatic, benzylic primary alcohols to aldehydes or carboxylic acids”.

Reply:We have checked the whole manuscript and corrected all the similar errors.

  1. write the complete name in the first appearance in the manuscript. The authors did follow this pattern. For example, in line 16 and 18.

Reply: Thank you. We have checked the whole manuscript and corrected all the similar errors. 

  1. Units are not correct in Figure 2, 3, and 4. Write it as for example “wavelength (nm)”.

Reply: All the units in the figures have been unified, for example, “Wavelength (nm)”, “Time (min)”, and “2q (degree)”. 

6.Correct all units. The units are not correct such as some where the unit of time is used mins and somewhere minutes.

Reply:All the units in the figures have been unified.

  1. The authors did not represent the 2θ values in Figure 2(c).

Reply: The marks of some characteristic diffraction peaks in XRD patterns will certainly help the identification. Because the crystal structures of MIL-101(Fe) and MIL-101(Cr) are well-studied (noted that these two MOFs have topologically isomorphic structures), the comparisons of diffraction patterns of synthesized MIL-101(Fe or Cr) and with simulated pattern are often used to depict the phase purity. For clarity purpose, the 2θ values were omitted.

  1. How can you confirm that the product is aldehydic. It may be possible that the alcoholic group was converted into carboxylic groups.

Reply: All the products in the manuscript have been identified by GC-MS. Indeed, the oxidation of primary alcohols by strong oxidants (e.g., KMnO4) often resulted in a mixture of aldehydes and carboxylic acid. To improve the selectivity, a large number of gentle catalytic systems have been developed, and TEMPO-based organocatalysts have been proved to be one of the most excellent catalysts. In our work, the IsoNTA-TEMPO/MIL-101(Fe) catalytic system show excellent catalytic activity and selectivity toward the oxidation of primary alcohols to their corresponding aldehydes.

  1. Title of article should be improved.

Reply: The tile of ‘Redox-Active Metal-Oganic Framework Nodes Boost Catalytic Activity of TEMPO-Mediated Aerobic Oxidation of Alcohols” covers the highlights of our work, that is, redox-active Fe(III) node boosts the catalytic activity of IsoNTA-TEMPO toward the oxidation of alcohols. For this reason, we think the tile is appropriate.

  1. It is very important to write the synthetic scheme which is missing in this manuscript.

Reply: The synthetic scheme of IsoNTA-TEMPO can be found in Scheme 1 in Supporting Information.

  1. Correct the molecular formulas. Some formulas are incorrect such as CrCl3•6H2O, FeCl3•6H2O

Reply: Thank you for your kind reminder. We have checked the whole manuscript and replaced the incorrect form with the correct ones as “CrCl3ž6H2O, FeCl3ž6H2O”.

  1. There are many previous works published for catalysis. The authors seem deliberately avoid those papers. The authors need to acknowledge the previous literature and compare their work with the similar ones and other systems in the literature and demonstrate their research outcomes in terms of advantages and disadvantages. Some of studies are given below need to cited;

https://doi.org/10.1016/j.cplett.2020.137645;

https://doi.org/10.1016/j.inoche.2022.109870;

https://doi.org/10.1039/D1RA09380K

Reply: We have carefully read the above-mentioned papers. Unfortunately, we found these papers are nice, but loosely related to our work. For example, the first one titled “Extraction of cobalt ions from aqueous solution by microgels for in-situ fabrication of cobalt nanoparticles to degrade toxic dyes: A two fold-environmental application” used free radical polymerization to build up microgels for degradation of toxic dyes. The synthetic method, chemicals or products involved in this paper seem irrelevant to our work. Similarly, the other two papers (three papers share many co-authors) deal with the degradation of dyes and reduction of nitroarenes by metal nanoparticles loaded microgels. We appreciated the hard work from Reviewers for improving our manuscript; Nevertheless, to meet the high standard of Molecules, we think it is important to make every citation in our manuscript supports our findings.

Reviewer 2 Report

Comments to the Author

The manuscript (molecules-2103424) titled ‘Redox-Active Metal-Oganic Framework Nodes Boost Catalytic Activity of TEMPO-Mediated Aerobic Oxidation of Alcohols’ has been carefully reviewed. The manuscript is a good contribution to aerobic oxidation of various alcohols to the corresponding aldehydes or ketones. The work elegantly applied the MOFs/TEMPO-IsoNTA catalysts by various characterization techniques. This context of article is suitable to the scope of this journal. The work is comprehensive and the data is well analyzed and presented. Overall, the study is deserved to be published in this journal, but minor revisions needed to be done as follows.

1. The first occurrence of TEMPO and TEMPO-IsoNTA in the manuscript should give the technical term.

2. In figure 1c, XRD patterns confirmed that the crystallinity of MIL-101(Fe) and MIL-101(Cr) materials had been changed. Please describe in detail and explain the reasons for the result caused by different materials.

3. The authors tried to evaluate the stability of MIL-101 (Fe)/TEMPO-IsoNTA, while the conversion was reduced in the third run, which was considered to be caused by the partially blocked pores of MIL-101(Fe). However, there is no satisfactory explanation to support the viewpoint. Please supply some convincing evidence, such as BET and FT-IR. Meanwhile, Fe3+/Fe2+ might be also changed during the reaction and it is recommended to supply the XPS results of the fresh and used catalysts.

4. Compared to MIL-101(Fe) and TEMPO-IsoNTA, MIL-101(Fe)/TEMPO-IsoNTA showed higher conversion in the various alcohols oxidation reactions. So please explain why it shows high catalytic performance. In addition, what are the reasons for the low activity of MIL-101(Cr)/TEMPO-IsoNTA?

5. The languages should be carefully checked and polished. For example, the figure 3d, “MIL-101(Fe)” should be changed to “MIL-101(Cr)”. In the line 267, the reaction “Fe2++NO2+2H+Fe3++NO+H2O” is wrong.

Author Response

Reviewer 2:

The manuscript (molecules-2103424) titled ‘Redox-Active Metal-Oganic Framework Nodes Boost Catalytic Activity of TEMPO-Mediated Aerobic Oxidation of Alcohols’ has been carefully reviewed. The manuscript is a good contribution to aerobic oxidation of various alcohols to the corresponding aldehydes or ketones. The work elegantly applied the MOFs/TEMPO-IsoNTA catalysts by various characterization techniques. This context of article is suitable to the scope of this journal. The work is comprehensive and the data is well analyzed and presented. Overall, the study is deserved to be published in this journal, but minor revisions needed to be done as follows.

Reply: We are thankful for the reviewer’s positive comments on our work and the agreement of publication in this journal. We humbly accept the valuable suggestions to revise the manuscript.

  1. The first occurrence of TEMPO and TEMPO-IsoNTA in the manuscript should give the technical term.

Reply: Thanks for your kind reminder. The full name of TEMPO-IsoNTA is N-(2,2,6,6-tetramethyl-1-piperidinyloxyl)isonicotinamide, and it has been added in the manuscript properly.

  1. In figure 1c, XRD patterns confirmed that the crystallinity of MIL-101(Fe) and MIL-101(Cr) materials had been changed. Please describe in detail and explain the reasons for the result caused by different materials.

Reply: MIL-101(Fe) and MIL-101(Cr) are two of the most important MOFs in MIL series and were first reported by Ferey’s group. According to their crystal structures, these two MOFs are known to have topologically isomorphic structures (meaning the same structural morphology, but with different compositions. Ref: J. Mater. Chem. A, 2021, 9, 22159). Typically, the comparisons of diffraction patterns of synthesized MIL-101(Fe or Cr) and with simulated pattern are often used to depict the phase purity. In our XRD data (Figure 2), despite the lower crystallinity of MIL-101(Fe) than that of MIL-101(Cr), the detailed comparisons indeed suggest the MIL-101(Fe) and MIL-101(Cr) have very similar patterns and is in good accordance with the simulated one. The slightly difference of the diffraction peaks between the MIL-101(Fe) and MIL-101(Cr) could rise from the defects, few trapped guest molecules (e.g. DMF).

  1. The authors tried to evaluate the stability of MIL-101 (Fe)/TEMPO-IsoNTA, while the conversion was reduced in the third run, which was considered to be caused by the partially blocked pores of MIL-101(Fe). However, there is no satisfactory explanation to support the viewpoint. Please supply some convincing evidence, such as BET and FT-IR. Meanwhile, Fe3+/Fe2+ might be also changed during the reaction and it is recommended to supply the XPS results of the fresh and used catalysts.

Reply: Thanks for your deep thinking. We have performed the XRD measurement of used MIL-101(Fe), and found the crystallinity of MOFs somewhat decreased. Moreover, XPS measurements of fresh and used MIL-101(Fe) were also carried out. As predicted by the reviewer, detailed XRD and XPS analysis suggested that the decreased crystallinity and imbalanced Fe3+/Fe2+ might be responsible for the decreased catalytic activity of MIL-101(Fe). The following discussion has been added to the manuscript.

“To understand the deceased catalytic activity of MIL-101(Fe), XRD pattern (Figure S7) and high-resolution Fe 2p XPS spectrum (Figure S8) of fresh and used MIL-101(Fe) were recorded. After use of three runs, some of the diffraction peaks (see Figure 7, marked by blue stars) of MIL-101(Fe) became less pronounced as compared to the fresh MIL-101(Fe), indicating decreased crystallinity of used MIL-101(Fe). In the XPS spectra of fresh MIL-101(Fe), Fe 2p3/2 and Fe 2p1/2 peaks associated with their satellite peaks (718.2 and 731.1 eV) were observed. The two sets of deconvoluted peaks located at (709.6 and 722.8 eV) and (711.1 and 724.8 eV) can be assigned to Fe2+ and Fe3+, respectively[36]. Similar deconvoluted peaks can be also obtained in the spectrum of used MIL-101(Fe), indicating the presence of Fe3+ and Fe2+. Detailed analysis of the mean relative areas suggested that the ratio of Fe3+/Fe2+ increased from 1:1 to 1.5:1 after use. The decreased crystallinity and imbalanced Fe3+/Fe2+ might be responsible for the decreased catalytic activity of MIL-101(Fe).”

  1. Compared to MIL-101(Fe) and TEMPO-IsoNTA, MIL-101(Fe)/TEMPO-IsoNTA showed higher conversion in the various alcohols oxidation reactions. So please explain why it shows high catalytic performance. In addition, what are the reasons for the low activity of MIL-101(Cr)/TEMPO-IsoNTA?

Reply: According to our experiments in Table 1, we believed that the superior catalytic activity of MIL-101(Fe)/TEMPO-IsoNTA hybrid system mainly attributed to the redox active Fe-oxo nodes in MIL-101(Fe) and a plausible catalytic mechanism was proposed in Figure 5. The low activity of MIL-101(Cr)/TEMPO-IsoNTA indeed was unexpected, since MIL-101(Cr) showed much high loading efficiency of TEMPO-IsoNTA. A possible reason might be that the strong Lewis acidity of Cr(III) allows coordination of TEMPO-IsoNTA by N-Ož→Cr(III), leading to decreased concentration of freely available N-Ož radical, thus lowering the catalytic activity of MIL-101(Cr)/TEMPO-IsoNTA. More experimental and theoretic work are certainly needed to gain insight into this unexpected effect. The following discussion has been added to the manuscript:

“A possible reason might be that the strong Lewis acidity of Cr(III) allows coordination of TEMPO-IsoNTA by N-Ož→Cr(III), leading to decreased concentration of freely available N-Ož radical, thus lowering the catalytic activity of MIL-101(Cr)/TEMPO-IsoNTA. Additional experimental and theoretic work are needed to fully understand this unexpected effect.”

  1. The languages should be carefully checked and polished. For example, the figure 3d, “MIL-101(Fe)” should be changed to “MIL-101(Cr)”. In the line 267, the reaction “Fe2++NO2+2H+Fe3++NO+H2O” is wrong.

Reply: Thank you for your careful reviewing. We have carefully checked the writing, language, format of the manuscript and polished the language further. According to previous reports (Ref. 23 and 35), the NO2 is able to oxide Fe2+ to Fe3+, accompanied by the production of H2O. This reaction formula can be also found in Ref. 35.

Reviewer 3 Report

The authors describe a catalytic system for oxidation of alcohols into aldehydes and ketones. After reading this paper, I had serious comments on the novelty and experimental nuances.
First, it is difficult to evaluate the novelty of this work. The MOFs used by the authors are already known and described in the references presented in this work. The description of the synthesis, XRD structure, SEM images have already been presented in previous papers.

The synthesis of organic TEMPO-IsoNTA has also been described previously. Consequently, for each component of the catalytic system there is already no novelty and this is a reproduction of the data.

In this context, it was important to establish the features of TEMPO-IsoNTA adsorption on the selected MOFs. However, the data presented do not allow us to claim the direct coordination of the organic ligand with the metal ions of MOFs.
It is worth noting that the spectra of the compound (TEMPO-IsoNTA) contain many impurities that are not integrated. The substance must be purified before further use. In this case it is a crude product without purification.

The catalytic processes also raise questions. Oxygen, TEMPO-IsoNTA, TBN in various combinations can act as oxidation initiators, so the proposed catalytic cycle is unlikely. TBN produces NO2 under harsher conditions. It is worth noting that in the case of ketones (not aldehydes) such catalytic systems can lead to the oxidation of methyl groups, which also raises great doubt that the selectivity in this case can reach such high values. Based on the above, I can't deduce any novelty for some field of research (synthesis of catalytic systems, catalysis) from the presented material.

On this view, I cannot recommend this work for publication in Molecules.

Author Response

Reviewer 3:

The authors describe a catalytic system for oxidation of alcohols into aldehydes and ketones. After reading this paper, I had serious comments on the novelty and experimental nuances. First, it is difficult to evaluate the novelty of this work. The MOFs used by the authors are already known and described in the references presented in this work. The description of the synthesis, XRD structure, SEM images have already been presented in previous papers. The synthesis of organic TEMPO-IsoNTA has also been described previously. Consequently, for each component of the catalytic system there is already no novelty and this is a reproduction of the data. In this context, it was important to establish the features of TEMPO-IsoNTA adsorption on the selected MOFs. However, the data presented do not allow us to claim the direct coordination of the organic ligand with the metal ions of MOFs. It is worth noting that the spectra of the compound (TEMPO-IsoNTA) contain many impurities that are not integrated. The substance must be purified before further use. In this case it is a crude product without purification. The catalytic processes also raise questions. Oxygen, TEMPO-IsoNTA, TBN in various combinations can act as oxidation initiators, so the proposed catalytic cycle is unlikely. TBN produces NO2 under harsher conditions. It is worth noting that in the case of ketones (not aldehydes) such catalytic systems can lead to the oxidation of methyl groups, which also raises great doubt that the selectivity in this case can reach such high values. Based on the above, I can't deduce any novelty for some field of research (synthesis of catalytic systems, catalysis) from the presented material.

On this view, I cannot recommend this work for publication in Molecules.

  1. First, it is difficult to evaluate the novelty of this work. The MOFs used by the authors are already known and described in the references presented in this work. The description of the synthesis, XRD structure, SEM images have already been presented in previous papers. The synthesis of organic TEMPO-IsoNTA has also been described previously. Consequently, for each component of the catalytic system there is already no novelty and this is a reproduction of the data.

Reply: We are sorry that the reviewer cannot catch the novelty of our work. Indeed, the MIL-101(Fe/Cr) were first reported by Ferey’s group and are two of the most important MOFs in MIL series. These two MOFs have been intensively studied by researchers over the world owing to their outstanding properties, including high thermal stability, open metal sites, high BET surface areas. However, exploration of their catalytic activity together with small organocatalysts are interesting and only few papers have been reported. Moreover, in our work, the TEMPO-IsoNTA was synthesized and used as an organocatalyst for aerobic oxidation of alcohols for the first time.

  1. ‘In this context, it was important to establish the features of TEMPO-IsoNTA adsorption on the selected MOFs. However, the data presented do not allow us to claim the direct coordination of the organic ligand with the metal ions of MOFs.’

Reply: Thank you for your useful information. The coordination of TEMPO-IsoNTA to MIL-101(Fe) or MIL-101(Cr) is indeed very important. According to general coordination chemistry, the pyridine group has high affinity to Fe3+ and Cr3+ owing to the presence of lone electron pair. Thus the coordination of TEMPO-IsoNTA to MIL-101(Fe) or MIL-101(Cr) through Py: →Fe(III) or Cr(III) is highly feasible. Moreover, a very similar molecule (ref. 37, JACS, 2009, 131, 7524), named [(S)-N-(pyridin-4-yl)-pyrrolidine-2-carboxamide, has been used to coordinate MIL-101(Cr) through Py: →Cr(III). In our case, because the TEMPO bears N-Ož radical, which also has potential to coordinate to Fe3+ or Cr3+ through N-Ož→Fe(III) or Cr(III), making the situation is more complicate. Our catalytic experiments suggested that, in the case of TEMPO-IsoNTA/MIL-101(Fe), the TEMPO-IsoNTA is likely bonded to Fe(III)-oxo node through Py: →Fe(III). However, in the case of IsoNTA/MIL-101(Cr), both Py: →Cr(III) and N-Ož→Cr(III) coordination modes are expected, because the N-Ož→Cr(III) will consume the freely available N-Ož radicals, and lowering the catalytic activity of the system, which is exactly observed in the experiments.

  1. ‘It is worth noting that the spectra of the compound (TEMPO-IsoNTA) contain many impurities that are not integrated. The substance must be purified before further use. In this case it is a crude product without purification.’

Reply: The newly synthesized TEMPO-IsoNTA was fully characterized by H NMR spectrum and C NMR spectra (Figure S1-S2) as well as by MALDI-TOF-MS (Figure S3). In the NMR spectra, there is almost no impurity. Please be aware that the additional peaks (taken as impurities by reviewer) arise from the reductant (phenylhydrazine, now has been marked by blue stars), since the radical specie is not suitable for NMR measurements.

  1. ‘The catalytic processes also raise questions. Oxygen, TEMPO-IsoNTA, TBN in various combinations can act as oxidation initiators, so the proposed catalytic cycle is unlikely. TBN produces NO2 under harsher conditions.’

Reply: Homogeneous or heterogeneous TEMPO mediated aerobic oxidation of alcohols has been intensively investigated, (ref. 23-26, and more refs: Chem. Commun., 2021, 57, 8897; Tetrahedron Lett., 2015, 56, 2768; Angew. Chem. Int. Ed. 2021, 60, 15686.). It has been well-established that the TBN could release NO under thermal conditions (e.g., 80 °C; see Ref. 23-26. And this temperature is typically regarded as a mild condition for aerobic oxidation of alcohols).

Round 2

Reviewer 1 Report

Accept

Author Response

Comments and Suggestions for Authors: Accept

Reply: Thank you very for the useful comments which help us to improve our manuscript.

Reviewer 3 Report

The authors have prepared a full response to the reviewer's comments. Revisions and corrections have been made to the text, which will be helpful to the reader. I can recommend this version of the article for publication after minor corrections.

Comments: Where does phenylhydrazine come from in ligand synthesis? For the authors' information, the presence of any substance in the spectrum refers to impurities. It is necessary to specify the purity of the target substance (in percent) based on the NMR data. In fact, phenylhydrazine acts as a reducing agent, which is harmful to oxidative processes and this fact can be taken into account when explaining the decrease in the catalytic reaction.

Author Response

The authors have prepared a full response to the reviewer's comments. Revisions and corrections have been made to the text, which will be helpful to the reader. I can recommend this version of the article for publication after minor corrections.

Comments: Where does come from in ligand synthesis? For the authors' information, the presence of any substance in the spectrum refers to impurities. It is necessary to specify the purity of the target substance (in percent) based on the NMR data. In fact, phenylhydrazine acts as a reducing agent, which is harmful to oxidative processes and this fact can be taken into account when explaining the decrease in the catalytic reaction.

Reply: Because of the paramagnetic nature of nitroxide radical, conventional NMR spectroscopy can not offer valuable structural information. However, one can reduce the nitroxide radical to its corresponding N-hydroxy amine product, for example, by phenylhydrazine (J. Org. Chem., 1975, 40, 3145.; Chem. Commun. 2009, 7, 836.; ChemCatChem 2014, 6 (8), 2419.) or ascorbic acid (Angew. Chem. Int. Ed. 2017, 56, 8892.) before NMR measurements. This is a well-established technique to explore the structure of nitroxide radical compounds. This means the phenylhydrazine was only introduced during the NMR measurements of TEMPO-IsoNTA. Nothing to do with phenylhydrazine neither in the synthetic procedure nor the catalytic reaction. To clearly state this information, the following text has been added into the manuscript:

“Because of the paramagnetic nature of nitroxide radical, conventional NMR spectroscopy can not offer valuable structural information. Therefore, phenylhydrazine was used to reduce the nitroxide radial to N-hydroxy amine before NMR measurements. In general, to a solution of TEMPO-IsoNTA (5 mg) in DMSOd6 (0.5 mL) in an NMR tube, 5 mL of phenylhydrazine was added. After 30 min, NMR spectrum of the corresponding N-hyroxyl amine was collected, and the purity of TEMPO-IsoNTA determined by NMR was higher than 97%.”
